# A Single Dose of Psilocybin Increases Synaptic Density and Decreases 5-HT_2A_ Receptor Density in the Pig Brain

**DOI:** 10.3390/ijms22020835

**Published:** 2021-01-15

**Authors:** Nakul Ravi Raval, Annette Johansen, Lene Lundgaard Donovan, Nídia Fernandez Ros, Brice Ozenne, Hanne Demant Hansen, Gitte Moos Knudsen

**Affiliations:** 1Neurobiology Research Unit, Copenhagen University Hospital Rigshospitalet, 2100 Copenhagen, Denmark; nakul.raval@nru.dk (N.R.R.); annette.johansen@nru.dk (A.J.); lenedonovan@nru.dk (L.L.D.); nidiafernandezros@gmail.com (N.F.R.); brice.ozenne@nru.dk (B.O.); hanne.d.hansen@nru.dk (H.D.H.); 2Faculty of Health and Medical Sciences, University of Copenhagen, 2200 Copenhagen, Denmark; 3Department of Public Health, Section of Biostatistics, Faculty of Health and Medical Sciences, University of Copenhagen, 1014 Copenhagen, Denmark; 4A. A. Martinos Center for Biomedical Imaging, Massachusetts General Hospital, Harvard Medical School, Charlestown, MA 02129, USA

**Keywords:** psilocybin, psychedelics, neuroplasticity, SV2A, 5-HT2A, depression, autoradiography, functional-selectivity

## Abstract

A single dose of psilocybin, a psychedelic and serotonin 2A receptor (5-HT_2A_R) agonist, may be associated with antidepressant effects. The mechanism behind its antidepressive action is unknown but could be linked to increased synaptogenesis and down-regulation of cerebral 5-HT_2A_R. Here, we investigate if a single psychedelic dose of psilocybin changes synaptic vesicle protein 2A (SV2A) and 5-HT_2A_R density in the pig brain. Twenty-four awake pigs received either 0.08 mg/kg psilocybin or saline intravenously. Twelve pigs (n = 6/intervention) were euthanized one day post-injection, while the remaining twelve pigs were euthanized seven days post-injection (n = 6/intervention). We performed autoradiography on hippocampus and prefrontal cortex (PFC) sections with [^3^H]UCB-J (SV2A), [^3^H]MDL100907 (5-HT_2A_R antagonist) and [^3^H]Cimbi-36 (5-HT_2A_R agonist). One day post psilocybin injection, we observed 4.42% higher hippocampal SV2A density and lowered hippocampal and PFC 5-HT_2A_R density (−15.21% to −50.19%). These differences were statistically significant in the hippocampus for all radioligands and in the PFC for [^3^H]Cimbi-36 only. Seven days post-intervention, there was still significantly higher SV2A density in the hippocampus (+9.24%) and the PFC (+6.10%), whereas there were no longer any differences in 5-HT_2A_R density. Our findings suggest that psilocybin causes increased persistent synaptogenesis and an acute decrease in 5-HT_2A_R density, which may play a role in psilocybin’s antidepressive effects.

## 1. Introduction

Serotonergic psychedelic drugs have for centuries been extensively used in religious practices and also recreationally [1]. Their neurobiological and behavioral effects in mammals are mediated through stimulation of the serotonin 2A receptor (5-HT_2A_R) as reviewed by Vollenweider et al. [2,3]. Upon ingestion of psilocybin, a tryptamine psychedelic [1], it quickly dephosphorylates to the active compound psilocin, which has a high affinity to 5-HT_2A_R, but also to other 5-HT receptors such as 5-HT_1A_R and 5-HT_2C_R [1,4,5].

Psychedelic stimulation of 5-HT_2A_R, a G-protein-coupled receptor (GPCR), has recently shown potential as an anxiolytic and antidepressant therapy. Some clinical studies suggest that a single dose of psilocybin rapidly and effectively relieves symptoms in depression and anxiety, with effects that persist long after the psychedelic experience [6,7,8,9]. Research in rodents suggests that psilocybin, lysergic acid diethylamide (LSD), 2,5-dimethoxy-4-iodoamphetamine (DOI), N,N-dimethyltryptamine (DMT), and alkaloids like harmine, tetrahydroharmine, and harmaline (present in ayahuasca) induce structural neuroplasticity and alter the expression of important proteins like VGLUT1, BDNF, kalirin-7 and MAP2 [10,11,12,13,14]. The mechanism behind these synaptic changes is hypothesized to be exerted via the 5-HT_2A_R pathway [10].

Changes in synaptic density in brain regions associated with emotional processing, i.e., the hippocampus and prefrontal cortex (PFC), may play a vital role in the pathophysiology of mood disorders, e.g., major depressive disorder. Both post-mortem human brain [15,16] and in vivo [17] studies in depressed individuals have shown a loss of synapses through the down-regulation of synaptic proteins and genes. Hence, upregulation of presynaptic proteins and an increase in synaptic density may be associated with the potential antidepressive effects of psychedelics.

Synaptic vesicle protein 2A (SV2A) is an integral 12-transmembrane domain glycoprotein expressed in synaptic vesicles throughout the brain [18], and SV2A density is thought to reflect presynaptic density [19]. The levetiracetam derivative UCB-J, which binds selectively to SV2A, has in its radiolabeled form been shown to correspond to synaptic density as measured with the well-characterized presynaptic protein synaptophysin [20,21,22].

Classical receptor binding assay studies have demonstrated that 5-HT_2A_R (and other GPCRs) exist in two affinity states, a high- and a low- affinity state [23,24,25]. The affinity states of the receptors are considered to represent different functional states of the receptor, high-affinity being functionally active (activation of G_αi1_-protein pathway) in contrast to the low-affinity state (activation of canonical G_αq/11_-protein pathway) [26]. Whereas 5-HT_2A_R antagonists bind to the total pool of 5-HT_2A_R, 5-HT_2A_R agonists bind to the high-affinity state GPCRs [27]. Stimulation of 5-HT_2A_R leads to rapid receptor internalization [28]. This endosomal internalization may lead to lysosomal degradation and down-regulation of 5-HT_2A_R, as extensively reviewed by Gray J.A. and Roth B.L [29].

In the present study, we hypothesize that a psychedelic dose of psilocybin increases presynaptic density, as reflected in SV2A protein levels in the pig brain. We also test the hypothesis that the availability of 5-HT_2A_R is decreased after agonist stimulation with psilocybin.

Using in vitro autoradiography, we measure SV2A and 5-HT_2A_R protein levels one and seven days post-injection of a single dose of psilocybin, known to induce 5-HT_2A_R associated behavioral changes corresponding to psychedelic effects, in healthy pigs [30]. We investigate brain effects one day after psilocybin administration because the days following a psychedelic experience may provide a therapeutic window to treat mood disorders [6,7]. A follow-up seven days after a psilocybin intervention was done because this is when depressive scores have been reported to be the lowest [6]. To investigate if potential psilocybin-induced reductions in 5-HT_2A_R are due to changes in the total receptor pool or confined to functionally active 5-HT_2A_Rs, we used both an antagonist ([^3^H]MDL100907) and an agonist radioligand ([^3^H]Cimbi-36) for autoradiography.

## 2. Results

### 2.1. SV2A Autoradiography

Figure 1a,b show the SV2A protein density as determined by [^3^H]UCB-J autoradiography in the hippocampus and the PFC. Compared to the saline-treated group, psilocybin treatment was associated with 4.42% higher SV2A in the hippocampus (*p* < 0.0001) one day after psilocybin injection and 9.24% higher SV2A in the hippocampus (*p* = 0.024) seven days after psilocybin (Figure 1a and Table 1). One day after psilocybin, there was no difference in PFC SV2A (Figure 2b and Table 1, 0.25%, *p* = 1), but seven days after psilocybin administration there was 6.10% higher SV2A in the PFC (*p* < 0.0001).

### 2.2. 5-HT_2A_R Antagonist Autoradiography:

Figure 2a,b show the 5-HT_2A_R density as measured with the antagonist radioligand [^3^H]MDL100907 in hippocampus and PFC. One day after the intervention, hippocampal 5-HT_2A_R density (Figure 2a and Table 1) was 29.60% lower (*p* < 0.0001) and PFC 5-HT_2A_R density was similar (−15.21%, *p* = 0.162, Figure 2b and Table 1) in the psilocybin-treated group than in the saline-treated group. Seven days after the psilocybin interventions, hippocampal and PFC 5-HT_2A_R density was not significantly different from the saline-treated animals.

### 2.3. 5-HT_2A_R Agonist Autoradiography

Figure 2c,d show the 5-HT_2A_R density as measured with the agonist radioligand [^3^H]Cimbi-36 in hippocampus and PFC. One day after psilocybin intervention, hippocampal 5-HT_2A_R/5-HT_2C_R density (Figure 2c and Table 1) was 43.39% lower (*p* = 0.013), and PFC 5-HT_2A_R density (Figure 2d and Table 1) was 50.19% lower (*p* < 0.0001) in the psilocybin-treated group than in the saline-treated group. With [^3^H]Cimbi-36, similar to [^3^H]MDL100907, 5-HT_2A_R density was not significantly different in the hippocampus and the PFC seven days after the psilocybin intervention compared to saline.

### 2.4. Antagonist vs. Agonist Radioligand for 5-HT_2A_R Density

We found a more pronounced reduction of 5-HT_2A_R density when measured with [^3^H]Cimbi-36 compared to [^3^H]MDL100907 one day after psilocybin intervention in the PFC. This difference of 41.26% was statistically significant at *p* = 0.033 (Table 1). We found no significant difference seven days after psilocybin intervention for either radioligand.

### 2.5. Plasma Psilocin

Plasma psilocin levels at euthanasia one and seven days after the psilocybin intervention were all below the detection limit.

## 3. Discussion

To the best of our knowledge, this is the first large-animal study to investigate how a single dose of psilocybin changes the key proteins SV2A and 5-HT_2A_R in brain regions involved in emotional processing. We find that a single dose of psilocybin increases the presynaptic marker, SV2A already after one day and that it remains higher seven days after. We also show a transient reduction in the hippocampus and PFC 5-HT_2A_R density; it is reduced one day after intervention but not seven days after.

The increase in synaptic marker SV2A may result from the stimulation of the 5-HT_2A_R, TrkB and mTOR-signaling pathways [10]. The activation of 5-HT_2A_R by DOI has been shown to induce a kalirin-7-dependent increase in dendritic spine size that may play a role in regulating structural plasticity in the cortex [14]. To understand the neurobiological basis of neuroplasticity and the implication of these changes, future proteomics studies must reveal which other proteins in these pathways are changed and the temporal evolvement of such changes. Our data support the notion of increased synaptogenesis following psychedelic exposure, which is hypothesized to underlie the antidepressant effects observed in humans: We find higher SV2A density in the hippocampus and the PFC, which are also regions where SV2A is reduced in patients with major depressive disorder [17]. Atypical antidepressants like ketamine are also associated with neuroplastic effects through proteins like cFos, pERK, and BDNF in the PFC and hippocampus in a social defect stress rodent model [31]. We propose that the increase in SV2A represents an increase in presynaptic density through the same pathways. The absence of psilocybin-associated changes in mRNA for cFos, pERK, and BDNF described in Donovan et al. [30] does not, however, exclude these pathways as instrumental mediators of psilocin’s effects on SV2A. This requires separate studies of protein levels or experiments where the pathways were interrupted. Together with other markers of neuroplasticity, increased levels of SV2A after intervention with a psychedelic drug adds to the scientific evidence that psychedelics enhance neuroplasticity, which may explain the mechanism of action of its antidepressant properties [32].

We have previously reported that 5-HT_2A_R mRNA expression is unaltered in the brains of these pigs [30]. It is, however, well known that brain 5-HT_2A_R mRNA expression does not correlate with 5-HT_2A_R protein content [33]. Our finding of a transient decrease in 5-HT_2A_R density, but not mRNA, one day after psilocybin is in line with results for the psychedelic substance DOI, where a significant difference in 5-HT_2A_R protein expression was not accompanied by a change in mRNA gene expression [34].

Compared to other protein-measuring techniques such as Western blot and immunohistochemistry, autoradiography provides an added advantage of measuring receptors in the functionally active vs. total receptor pool by the use of agonist or antagonist radioligands, respectively. The use of adjacent brain sections provides the ability to directly compare receptors. We find a statistically significant reduction in 5-HT_2A_R density in the PFC one day after psilocybin administration when measuring with [^3^H]Cimbi-36 compared to [^3^H]MDL100907. The difference between the two radioligands offers circumstantial evidence of the differential binding of antagonists versus agonists, at least when it comes to the PFC. More caution should be exerted when comparing the radioligands in the hippocampus because the hippocampus has high levels of 5-HT_2C_R, with a density similar to 5-HT_2A_R [35,36] and [^3^H]Cimbi-36 also has affinity to 5-HT_2C_R [37,38]. That is, we cannot exclude the possibility that some of the observed reduction in [^3^H]Cimbi-36 in the hippocampus could be due to a down-regulation of 5-HT_2C_R. It could be a concern that the reduction in 5-HT_2A_R one day after psilocybin was due to partial blocking by residual psilocin. However, plasma psilocin levels at euthanasia one day after psilocybin administration were under the detection limit in all animals.

The fraction of functional 5-HT_2A_R has to some extent been assessed in vivo in non-human primates [38] and humans [39] using [^11^C]Cimbi-36 as an agonist and [^11^C]MDL100907 or [^18^F]altanserin as antagonist radioligands, but a more precise estimate is difficult in vivo because of missing information about the free fraction of the radioligand and the radioligand affinity. A little unexpectedly, B_max_ did not differ substantially between agonist and antagonist radioligands in our study, but it should be kept in mind that uncertainties in the determination of specific activities are reflected in the calculation of B_max_. Functional receptors can also be measured with [^35^S]guanosine triphosphate (GTP) γS binding stimulation mediated with DOI followed by immunoprecipitation with specific antibodies, while the complex is captured with protein A-polyvinyl toluene scintillation proximity assay. When this approach is made on post-mortem brain tissue from patients with schizophrenia, the canonical G_αq/11_-protein pathway of 5-HT_2A_Rs is found to be unaltered in the PFC, whereas the pro-hallucinogenic G_αi1_-protein pathway is functionally overactive in the PFC [40,41]. GTPγS binding assay may be more sensitive to the measurement of functional receptors and could generate an outcome that was more straight-forward to interpret.

It is already well-known that a transient 5-HT_2A_R down-regulation occurs upon agonist stimulation, followed by a return to baseline [42]. We have previously found that the 5-HT_2A_R binding has normalized seven days after healthy individuals take a single psychedelic dose of psilocybin [43]. To what extent the transient down-regulation of 5-HT_2A_R is a prerequisite for boosting the formation of new synapses is intriguing and should be examined in future studies.

Some limitations of the study should also be mentioned. We chose to investigate only two time-points and selected a few highly relevant proteins in two relevant brain regions. It would be interesting to investigate whether the synaptic density increases further beyond one week, and for how long it is maintained. Although we cannot be certain that our findings translate to humans that consume a single dose of psilocybin, the SV2A density in the hippocampus and the PFC in the saline treated pigs is in the same range as that reported in post-mortem human and non-human primates by Varnäs et al. [22]. The changes in SV2A and 5-HT_2A_R were seen in healthy pigs; it might also be relevant to investigate changes in a psychosocial chronic-stress pig model [44]. Further, to ensure that the pigs received a well-defined dose of psilocybin, we chose to administer the drug intravenously rather than perorally. This differs from the typical approach in patients. Despite the faster pharmacokinetics after intravenous administration, the dose and administration route result in the same 5-HT_2A_R occupancy as in humans that take it perorally [30,43].

## 4. Materials and Methods

### 4.1. Animals and Drug Dosage

The brain tissue was retrieved from pigs entering a previously published study [30] where more details are described. Briefly, female Danish slaughter pigs (Yorkshire × Duroc × Landrace) weighing around 20 kg (approximately nine weeks old) were used in the study. The animals were sourced from a local farm and allowed to acclimatize for at least one week before the start of the experiment. The animals were housed in individual pens with an enriched environment on a 12-h light/dark cycle, with free access to water, and weight-adjusted food twice daily. The welfare of the animals was assessed daily. After arriving in the stables, animals were trained for up to a week to allow for handling by humans.

Donovan et al. [30] identified which dose of psilocybin to give to make it comparable to a dose that elicits psychedelic effects in humans. This was based on behavioral response (headshakes, hindlimb scratches, and rubbing against the pen wall) and on Positron emission tomography (PET) studies of the 5-HT_2A_R occupancy using the agonist radioligand [^11^C]Cimbi-36; 67% 5-HT_2A_R occupancy will elicit psychedelic effects in humans [43]. Intravenous injection of psilocybin was given in an ear vein catheter to the awake pigs, and the animals were under no form of external stress during the experiment. At the time of euthanasia, blood was drawn for the measurement of plasma psilocin levels, which were measured by ultra-high-performance liquid chromatography coupled to tandem mass spectrometry as previously described [43].

### 4.2. Ethical Statement

All animal experiments conformed to the European Commission’s Directive 2010/63/EU and the ARRIVE guidelines. The Danish Council of Animal Ethics had approved all procedures (Journal no. 2016-15-0201-01149).

### 4.3. Study Design

Figure 3 shows the overall design of the study. Twenty-four awake pigs were given an intravenous dose of either 0.08 mg/kg psilocybin (n = 12) or saline (n = 12) through an ear vein catheter. Half of the animals in each group were euthanized one day post-injection (n = 6/intervention). The remaining 12 animals were euthanized seven days post-injection (n = 6/intervention). That is, the animals were divided into four groups: Saline: 1 day, Psilocybin: 1 day, Saline: 7 days and Psilocybin: 7 days (Figure 3). The extracted brains were snap-frozen and stored at −80 °C. From one hemisphere, 20 µm thick frozen sections were sliced on a cryostat (Leica CM1800, Leica Biosystems, Buffalo Grove, IL, USA) from the PFC and the hippocampus and mounted on Superfrost Plus™ adhesion microscope slides. Sections were stored at −20 °C for the remaining period of the study.

### 4.4. Autoradiography

Radioligands used for autoradiography included SV2A imaging with [^3^H]UCB-J (UCB pharma, Brussels, Belgium, specific activity 14 Ci/mmol or Pharmaron Ltd., Hoddesdon, UK, specific activity 28 Ci/mmol). [^3^H]MDL100907 (ViTrax, Placentia, CA, USA, specific activity 56 Ci/mmol) was used as an antagonist radioligand for 5-HT_2A_R and [^3^H]Cimbi-36 (kindly provided by Prof. Dr C. Halldin, Department of Neuroscience, Karolinska Institute, Stockholm, Sweden, specific activity 53 Ci/mmol,) as an agonist radioligand for 5-HT_2A_R/5-HT_2c_R. Radio-Thin-Layered-Chromatography (R-TLC) was performed for all radioligands to measure the radiochemical purity (RCP) and integrity of the parent compound. The mobile phase for [^3^H]UCB-J R-TLC was Acetonitrile:Ammonium formate [25:75] (0.1 M, with 0.5% AcOH, pH 4.2). The mobile phase for [^3^H]MDL100907 R-TLC was Chloroform:Methanol:Ammonia solution [90:9:1]. The mobile phase for [^3^H]Cimbi-36 R-TLC was Chloroform:Methanol:Triethylamine [94:5:1]. [^3^H]UCB-J and [^3^H]MDL100907 had high RCP (96–98%) while [^3^H]Cimbi-36 had an RCP of 52–57%. Radioactivity was corrected for RCP of [^3^H]Cimbi-36 after TLC.

Sections were thawed to room temperature for 30–45 min before prewashing twice for 10 min in 50 mM Tris-HCl pre-incubation buffer set to 7.4 pH containing 0.5% bovine serum albumin (BSA) for [^3^H]UCB-J or 0.01% ascorbic acid, 4 nM CaCl_2_ and 0.1% BSA for [^3^H]MDL100907 and [^3^H]Cimbi-36.

For SV2A, the sections were incubated in assay buffer containing 60 nM [^3^H]UCB-J in 50 mM Tris-HCl buffer containing 5 mM MgCl_2_, 2 mM EGTA and 0.5% BSA (pH 7.4) for 1 h. Incubation was terminated by three 5-min washes with 4 °C pre-incubation buffer followed by a rapid rinse in 4 °C deionized H_2_O (dH_2_O). For 5-HT_2A_R, sections were incubated in assay buffer containing 3 nM [^3^H]MDL100907 or 1 nM [^3^H]Cimbi-36 in 50 mM Tris-HCl containing 0.01% ascorbic acid, 4 nM CaCl_2_ and 0.1% BSA (pH 7.4) for 1 h. Incubation was terminated by two 10-min washes in ice-cold pre-incubation buffer followed by a rapid rinse in ice-cold dH_2_O.

The assay buffer concentration of the respective radioligands was determined using 4–5 × K_D_ values (Appendix A: Figure A1) to determine B_max_ values in the section. After washing, the slides were rapidly air-dried and fixated in a paraformaldehyde vapor chamber overnight in cold storage (4 °C). The next day, the samples were moved to an exicator for 45–60 min to remove any excess moisture and then placed in a cassette for autoradiography with tritium sensitive image plates (BAS-IP TR2040, Science Imaging Scandinavia AB, Nacka, Sweden) along with radioactive tritium standards (RPA510, Amersham Bioscience, GE Healthcare, Chicago, IL, USA) (Figure 4). The image plates were exposed for seven days. After the exposure, the image plates were read using a Fujifilm BAS 1000 scanner (Fujifilm Europe, GmbH, Duesseldorf, Germany). Calibration, quantification and data evaluation were done using ImageJ software (NIH Image, Bethesda, MD, USA) [45]. The regions of interest were hand-drawn or drawn using the wand tool and visually inspected after automated delineation, as shown in Figure 4. The four-parameter general curve fit (David Rodbard, NIH) of decay corrected tritium standards was used to convert mean pixel density (grayscale) to nCi/mg tissue equivalent (TE). Total binding was determined in the hippocampal and cortical grey matter while non-specific binding was determined in the white matter on the same slides. Finally, the decay-corrected specific activity of the representative radioligand was used to convert nCi/mg TE to fmol/mg TE. Specific binding was calculated as the difference between total binding and non-specific binding. All experiments were performed in triplicates, and experimenters were blinded.

### 4.5. Statistical Analyses

The data were analyzed using R (v. 4.0.3; R core team, Vienna, Austria), while GraphPad Prism (v. 9.0.0; GraphPad Software, San Diego, CA, USA) was used for data visualization. Comparisons between group means ($\;\bar{X}$) for the respective radioligands (Equations (1) and (2)) were done using a permutation test (with 1000 permutations) on log-transformed values and adjusted for multiple comparisons (overtime, radioligand, and brain regions: 12 tests) using the Holm method. Comparison between the 5-HT_2A_R radioligands, [^3^H]Cimbi-36 and [^3^H]MDL100907, was performed for the PFC using a permutation test (with 1000 permutations) on log-transformed values of treatment effect at one day and seven days (Equations (3) and (4)).
(1)$$X\;(\mathrm{Psilocybin}:1\;\mathrm{day})-(\bar{X})\;(\mathrm{Saline}:1\;\mathrm{day})$$
(2)$$X\;(\mathrm{Psilocybin}:7\;\mathrm{day})-(\bar{X})\;(\mathrm{Saline}:7\;\mathrm{day})$$
(3)$${[}^{3}H]\mathrm{Cimbi}-36((\bar{X})\;(\mathrm{Psilocybin}:1\;\mathrm{day})-(\bar{X})\;(\mathrm{Saline}:1\;\mathrm{day}))-{[}^{3}H]\mathrm{MDL}100907((\bar{X})\;(\mathrm{Psilocybin}:1\;\mathrm{day})-(\bar{X})\;(\mathrm{Saline}:1\;\mathrm{day}))$$
(4)$${[}^{3}H]\mathrm{Cimbi}-36((\bar{X})\;(\mathrm{Psilocybin}:7\;\mathrm{days})-(\bar{X})\;(\mathrm{Saline}:7\;\mathrm{days}))-{[}^{3}H]\mathrm{MDL}100907((\bar{X})\;(\mathrm{Psilocybin}:7\;\mathrm{days})-(\bar{X})\;(\mathrm{Saline}:7\;\mathrm{days}))$$

## Figures and Tables

**Figure 1 ijms-22-00835-f001:**
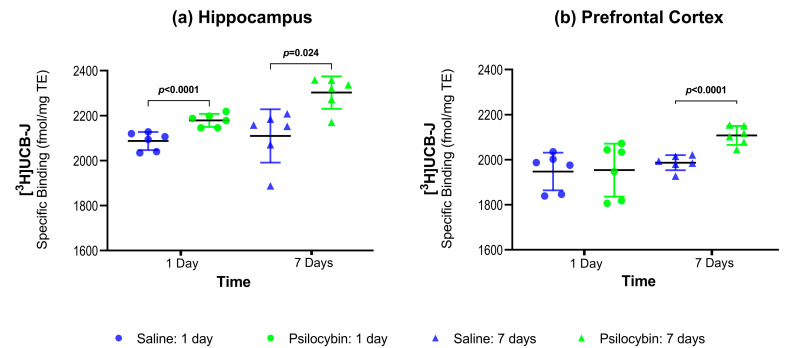
Group-wise comparisons of synaptic vesicle protein 2A (SV2A) density (mean ± SD) in the hippocampus (**a**) and prefrontal cortex (PFC) (**b**) as measured with [^3^H]UCB-J autoradiography.

**Figure 2 ijms-22-00835-f002:**
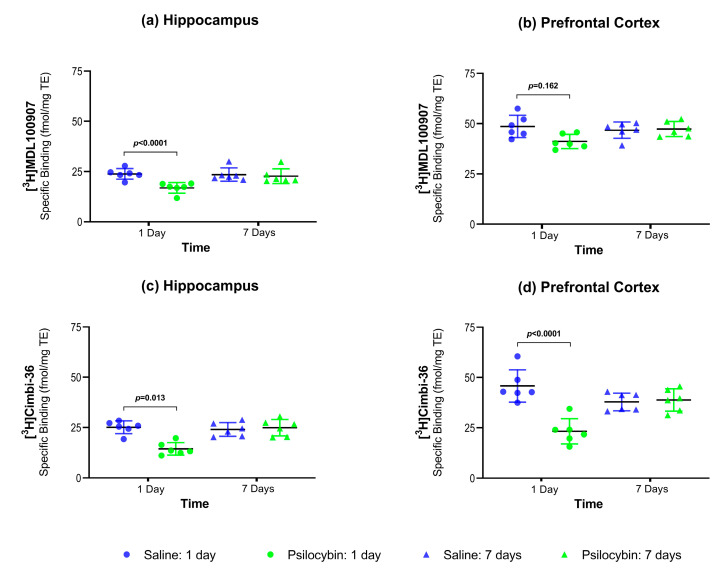
Group-wise comparison of 5-HT_2A_R density (mean ± SD) as measured with [^3^H]MDL100907 and [^3^H]Cimbi-36 in the hippocampus (**a**,**c**) and PFC (**b**,**d**) using autoradiography.

**Figure 3 ijms-22-00835-f003:**
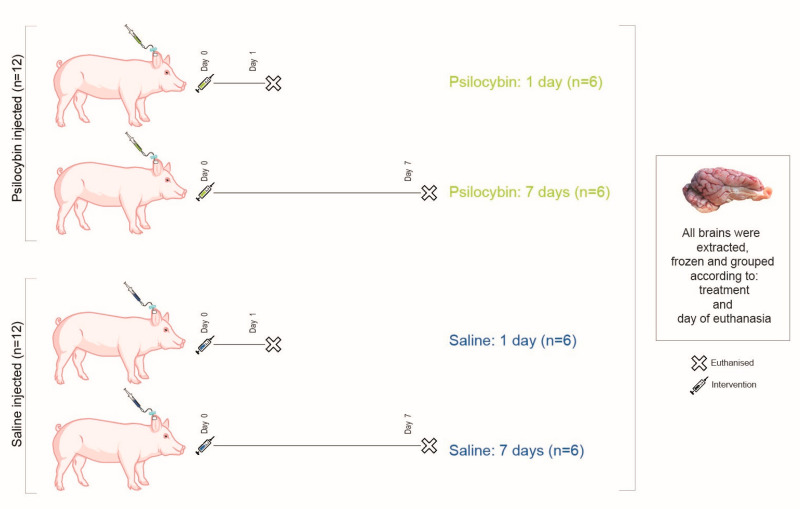
Study design. Twenty-four pigs received an intravenous dose of either 0.08 mg/kg psilocybin or saline. Six pigs from each type of intervention were euthanized one day or seven days post-injection. The pigs were divided into four groups, as depicted in the figure.

**Figure 4 ijms-22-00835-f004:**
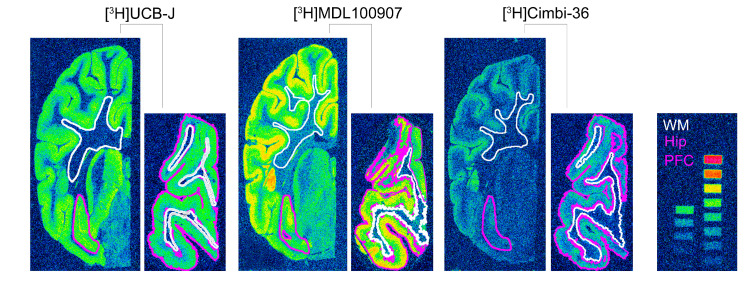
Representative autoradiograms of the radioligands used in this study. Half hemisphere sections of the hippocampus (larger coronal sections) and PFC (smaller coronal sections) from the same animal belonging to the Saline: 1 day group. The color-coded lines show representations of the region of interest that were either hand-drawn or drawn using the wand tool in ImageJ and visually inspected. The figure also shows the radioactive standards used in the study; approximately 16 nCi/mg to 0.2 nCi/mg decay corrected to the time of experiment and day of exposure.

**Table 1 ijms-22-00835-t001:** Group-wise summary of statistical tests performed for each radioligand. All tests show the adjusted *p*-values using the Holm method. NA (not applicable) indicates test was not performed.

	Hippocampus		Prefrontal Cortex	
	1 day	7 Days	1 Day	7 Days
[^3^H]UCB-J (psilocybin vs. saline)	+4.42% (*p* < 0.0001)	+9.24% (*p* = 0.024)	+0.25% (*p* = 1)	+6.10% (*p* < 0.0001)
[^3^H]MDL100907 (psilocybin vs. saline)	−29.60% (*p* < 0.0001)	−3.58% (*p* = 1)	−15.21% (*p* = 0.162)	+1.32% (*p* = 1)
[^3^H]Cimbi-36 (psilocybin vs. saline)	−43.39% (*p* = 0.013)	+3.31% (*p* = 1)	−50.19% (*p* < 0.0001)	+2.23% (*p* = 1)
[^3^H]MDL100907 vs. [^3^H]Cimbi-36	NA	NA	−41.26% (*p* = 0.033)	+0.90% (*p* = 0.921)

## Data Availability

The data presented in this study are available on request from the corresponding author. The data are not publicly available due to other on-going studies.

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
