# Peer review of "A Single Dose of Psilocybin Increases Synaptic Density and Decreases 5-HT2A Receptor Density in the Pig Brain"

_ijms, 2021, doi:10.3390/ijms22020835_

Round 1
Reviewer 1 Report
Knudsen and colleagues have recently focused their research on psilocin both in humans and in pigs.
This research follows the study by Donovan et al., 2020 in pigs. Here the authors investigated the effect of a single psychedelic dose of psilocybin on synaptic vesicle protein 2A (SV2A) and 5-HT2AR density. They found one day post psilocybin injection, higher hippocampal and lowered hippocampal SV2A density and PFC 5-HT2AR density. Seven days post-intervention, there was still significantly higher SV2A density in hippocampus and PFC whereas there were no longer any differences in 5-HT2AR density. The authors suggest that psilocybin’s antidepressive actions are linked to increased persistent synaptogenesis and possibly also to an acute decrease in 5-HT2AR density.
The research is well conducted, the results are interesting and the experimental approach is correct.
I have only some minor consideration for the authors.
The most important point is the fact that the authors both in the abstract and in the discussion assert that these changes seen are correlated with the antidepressant effect mediated by psilocybin. This is not the case, considering that the changes observed are found in normal pigs. Different experiments should be performed to arrive at such conclusions.
From what we know these changes maybe not be causative. Therefore the authors should avoid overestimation of their results (i.e., do not use “link to” depression in the abstract).
Define CPCR the first time
Update ref 29
Plasma psilocin results are mentioned and not shown. Please, specify if these data are original.
Is Psilocybin occupancy of [11C]Cimbi-36 taken from ref 29 or original?
Do the pigs experimented here the same of ref 29?
To determine the degree of 5-HT2C receptor occupancy, the author should use [3H]mesulergine binding for comparison.
Bryan Roth has recently published groundbreaking research that used cryogenic electron microscopy (cryo-EM) to illustrate precisely how psychedelic compounds bind to the 5-HT2A serotonin receptor complex. These findings (Kim et al., 2020) were published in the journal Cell. Please, include this in the discussion.
Author Response
We sincerely thank you for your valuable time and comment. This review indeed helps us to improve the manuscript. Here, we further address and answer your comments.
- The most important point is the fact that the authors both in the abstract and in the discussion assert that these changes seen are correlated with the antidepressant effect mediated by psilocybin. This is not the case, considering that the changes observed are found in normal pigs. Different experiments should be performed to arrive at such conclusions.
- From what we know these changes maybe not be causative. Therefore the authors should avoid overestimation of their results (i.e., do not use “link to” depression in the abstract).
We agree with the reviewer and have toned our interpretation down in the abstract and the discussion:
Abstract, page 1, line 29:
“Our findings suggest that psilocybin causes increased persistent synaptogenesis and an acute decrease in 5-HT2AR density, which may play a role for psilocybin’s antidepressive effects.”
Discussion, page 5, line 141:
“Together with other markers of neuroplasticity, increased levels of SV2A after intervention with a psychedelic drug adds to the scientific evidence that psychedelics enhance neuroplasticity which may explain the mechanism of action of its antidepressant properties [32].”
- Define CPCR the first time
GPCR is defined and abbreviated in the introduction on page 2, line 43.
- Update ref 29
The reference (now 30) has now been updated to the latest version of the published article.
- Plasma psilocin results are mentioned and not shown. Please, specify if these data are original.
The plasma psilocin results are original in the sense that they were not included in Donovan et al. [reference 30], and they are mentioned in the Results, page 4, line 118-120. Since they all are under the detection limit, they are not presented in a tabular form.
- Is Psilocybin occupancy of [11C]Cimbi-36 taken from ref 29 or original?
- Do the pigs experimented here the same of ref 29?
As we mention in the Materials and Methods, page 6, line 191 and 199, the pig brain tissue was retrieved and used from the previous study, Donovan et al. [reference 30]. It is also mentioned that the PET-based occupancy measure is from the same study.
- To determine the degree of 5-HT2C receptor occupancy, the author should use [3H]mesulergine binding for comparison.
We did not intend to measure 5-HT2CR density because it has been firmly established that it is the stimulation of the 5-HT2AR that leads to the psychedelic experience. That is, we would deem the determination of 5-HT2CR density out of the scope of this study.
- Bryan Roth has recently published groundbreaking research that used cryogenic electron microscopy (cryo-EM) to illustrate precisely how psychedelic compounds bind to the 5-HT2A serotonin receptor complex. These findings (Kim et al., 2020) were published in the journal Cell. Please, include this in the discussion.
We are well aware of Prof. Bryan Roth and colleagues’ brilliant study on the structure of psychedelic bound 5-HT2AR complex, but unfortunately, we cannot relate those molecular observations directly to the discussion of our ex vivo data.
Reviewer 2 Report
The manuscript entitled, "A single dose of psilocybin increases synaptic density and decreases 5-HT2A receptor density in the pig brain", is of general interest and of appropriate level.
I have minor comments:
Comment 1:
Line 155
"It could be a concern that the reduction in 5-HT2AR one day after psilocybin was due to partial blocking by residual psilocin. However, plasma psilocin levels at euthanasia one day after was under the detection limit in all animals"
Would it not make more sense to know the level of psilocin in the brain?
Comment 2:
In my view in the discussion a possible explanation for the down-regulation of 5-HT2AR and an increased in the SV2A density is missing. Could the authors elaborate more on this hypothesis?
Author Response
We sincerely thank you for your valuable time and comment. Your concise review has helped us to improve the manuscript. Here, we further address and answer your comments.
- Would it not make more sense to know the level of psilocin in the brain?
Thank you for this excellent comment; it may indeed be more relevant to directly measure psilocin levels in the target tissue. However, we are not so concerned about a potential disequilibrium between plasma and brain psilocin since 1) data from humans show an excellent correlation between plasma psilocin and 5-HT2AR occupancy over time (Madsen et al. 2019) and 2) Based on the plasma psilocin pharmacokinetics after iv injection of psilocybin in the pig where the half-life is 35-40 min (Donovan et al. 2021) the estimated plasma level of psilocin 24 hours after injection is negligible 3) No behavioural changes were observed in the pigs 2 hours after psilocybin injection, supporting that brain levels of psilocin were equally rapidly reduced.
- In my view in the discussion a possible explanation for the down-regulation of 5-HT2AR and an increased in the SV2A density is missing. Could the authors elaborate more on this hypothesis?
We agree that identifying the mechanism behind our observations is highly interesting, but we can only speculate. We already briefly discuss the agonist-mediated 5-HT2AR decrease in the light of previous findings of internalization by Roth et al. on page 5, line 126 (paragraph 2 of discussion). Concerning the mechanism for synaptic density, we have added the following to the discussion:
Discussion, page 4, Line 129-130
“Activation of 5-HT2AR by DOI has shown to induces a kalirin-7 dependent increase in spine size that may play a role in regulating structural plasticity in the cortex.”
Discussion, page 5, Line 136-142
“We propose that the increase in SV2A represents an increase in presynaptic density through the same pathways. The absence of psilocybin-associated changes in mRNA for cFos, pERK and BDNF described in Donovan et al. [30] does not, however, exclude these pathways as instrumental mediators of psilocin’s effects on SV2A. This requires separate studies of protein levels or experiments where the pathways were interrupted.”